# Intestinal Insights: The Gut Microbiome’s Role in Atherosclerotic Disease: A Narrative Review

**DOI:** 10.3390/microorganisms12112341

**Published:** 2024-11-16

**Authors:** Luana Alexandrescu, Adrian Paul Suceveanu, Alina Mihaela Stanigut, Doina Ecaterina Tofolean, Ani Docu Axelerad, Ionut Eduard Iordache, Alexandra Herlo, Andreea Nelson Twakor, Alina Doina Nicoara, Cristina Tocia, Andrei Dumitru, Eugen Dumitru, Laura Maria Condur, Cristian Florentin Aftenie, Ioan Tiberiu Tofolean

**Affiliations:** 1Gastroenterology Department, “Sf. Apostol Andrei” Emergency County Hospital, 145 Tomis Blvd., 900591 Constanta, Romania; alexandrescu_l@yahoo.com (L.A.); cristina.tocia@yahoo.com (C.T.); dr.andreidumitru@gmail.com (A.D.); eugen.dumitru@yahoo.com (E.D.); tofoleanioan@yahoo.com (I.T.T.); 2Medicine Faculty, “Ovidius” University of Constanta, 1 Universitatii Street, 900470 Constanta, Romania; alina.stanigut@365.univ-ovidius.ro (A.M.S.); tofoleandoina@yahoo.com (D.E.T.); axelerad.docu@365.univ-ovidius.ro (A.D.A.); alina.nicoara@365.univ-ovidius.ro (A.D.N.); lauracondur@yahoo.com (L.M.C.); afteniecristian@gmail.com (C.F.A.); 3Nephrology Department, “Sf. Apostol Andrei” Emergency County Hospital, 145 Tomis Blvd., 900591 Constanta, Romania; 4Pneumology Department, Sf. Apostol Andrei” Emergency County Hospital, 145 Tomis Blvd., 900591 Constanta, Romania; 5Department of General Surgery, “Sf. Apostol Andrei” Emergency County Hospital, 145 Tomis Blvd., 900591 Constanta, Romania; eduardiordache@yahoo.com; 6Department XIII, Discipline of Infectious Diseases, “Victor Babes” University of Medicine and Pharmacy Timisoara, 2 Eftimie Murgu Square, 300041 Timisoara, Romania; alexandra.mocanu@umft.ro; 7Internal Medicine Department, Sf. Apostol Andrei” Emergency County Hospital, 145 Tomis Blvd., 900591 Constanta, Romania; andreea.purcaru@365.univ-ovidius.ro; 8Academy of Romanian Scientist, 3 Ilfov Street, 050044 Bucharest, Romania

**Keywords:** atherosclerosis, gut metagenome, inflammation, intestinal microbiota, probiotics, prebiotics

## Abstract

Recent advances have highlighted the gut microbiota as a significant contributor to the development and progression of atherosclerosis, which is an inflammatory cardiovascular disease (CVD) characterized by plaque buildup within arterial walls. The gut microbiota, consisting of a diverse collection of microorganisms, impacts the host’s metabolism, immune responses, and lipid processing, all of which contribute to atherosclerosis. This review explores the complex mechanisms through which gut dysbiosis promotes atherogenesis. We emphasize the potential of integrating microbiota modulation with traditional cardiovascular care, offering a holistic approach to managing atherosclerosis. Important pathways involve the translocation of inflammatory microbial components, modulation of lipid metabolism through metabolites such as trimethylamine-N-oxide (TMAO), and the production of short-chain fatty acids (SCFAs) that influence vascular health. Studies reveal distinct microbial profiles in atherosclerosis patients, with increased pathogenic bacteria (*Megamonas*, *Veillonella*, *Streptococcus*) and reduced anti-inflammatory genera (*Bifidobacterium*, *Roseburia*), highlighting the potential of these profiles as biomarkers and therapeutic targets. Probiotics are live microorganisms that have health benefits on the host. Prebiotics are non-digestible dietary fibers that stimulate the growth and activity of beneficial gut bacteria. Interventions targeting microbiota, such as probiotics, prebiotics, dietary modifications, and faecal microbiota transplantation (FMT), present effective approaches for restoring microbial equilibrium and justifying cardiovascular risk. Future research should focus on longitudinal, multi-omics studies to clarify causal links and refine therapeutic applications.

## 1. Introduction

The gut metagenome has been identified as an environmental factor affecting adiposity and obesity through the modulation of host lipid metabolism [1]. It serves as a source of inflammatory chemicals, including lipopolysaccharide (LPS) and peptidoglycan, which may contribute to metabolic diseases [2]. The gut microbiota, a dynamic and diverse community of trillions of microorganisms, plays a significant role in human health, influencing numerous physiological processes, including metabolism, immune modulation, and even brain function [3]. The gut microbiota consists of six phyla: *Firmicutes*, *Bacteroidetes*, *Actinobacteria*, *Proteobacteria*, *Fusobacteria*, and *Verrucomicrobia*, with *Firmicutes* and *Bacteroidetes* being the predominant types. The fungi most frequently examined in gut microbiota studies include *Candida*, *Saccharomyces*, *Malassezia*, and *Cladosporium*. The human gut microbiota includes not only bacteria and fungi but also viruses, phages, and archaea, predominantly *M. smithii* [4,5]. Dysbiosis, or the imbalance of microbial species within the gut, has been increasingly linked to metabolic and inflammatory diseases, ranging from obesity and type 2 diabetes to cardiovascular diseases like atherosclerosis [6,7]. Researchers are increasingly acknowledging the “gut–heart axis” as a critical focus for understanding the role of gut microbiota in the pathogenesis of atherosclerosis [8].

Atherosclerosis is a chronic inflammatory condition resulting from dysregulation of lipid metabolism and an inappropriate immune response, characterized by the buildup of cholesterol-rich macrophages within the arterial wall [9]. The gut microbiota metabolizes the dietary lipid phosphatidylcholine into trimethylamine, which induces atherosclerosis and inflammation [10]. Additionally, levels of choline, trimethylamine N-oxide, and betaine have been identified as predictors of CVD risk in humans [11]. Lv et al. utilized pyrosequencing of the 16S rRNA gene, revealing that atherosclerotic plaques harbor bacterial DNA with phylotypes prevalent in the gut microbiota and that the quantity of bacterial DNA in the plaques is linked with inflammation [12].

Furthermore, the gut microbiota influences lipid metabolism by regulating the absorption and processing of dietary fats and cholesterol [13]. Specific microbial communities facilitate lipid accumulation in the arterial wall, thereby advancing the development of atherosclerotic plaques. Short-chain fatty acids, specifically acetate, propionate, and butyrate, are generated by gut bacteria via the fermentation of dietary fibers, representing a significant connection in this relationship [14]. SCFAs display both pro-inflammatory and anti-inflammatory effects on vascular health. Leukocytes must undergo diapedesis to reach the damage site [15]. Adhesion molecules help this process. For example, propionate demonstrates anti-inflammatory properties that may offer protection against atherosclerosis [14]. Conversely, acetate may enhance cholesterol production, exacerbating lipid buildup in arterial walls [15].

Studies utilizing high-throughput sequencing have demonstrated that patients with symptomatic atherosclerosis often display an increase in pathogenic genera, such as *Megamonas*, *Streptococcus*, and *Veillonella* [16]. They also show a reduction in beneficial bacteria, including *Bifidobacterium* and *Roseburia* [16]. The compositional shifts may create an inflammatory environment that promotes atherosclerotic changes. This shows the potential of gut microbiota as both a biomarker and therapeutic target for CVD [17].

Microbiota-targeting methods face many challenges when used in clinical practice. Due to study designs, patient groups, intervention methods, and clinical trial outcomes are inconsistent, which is a major issue. The developing regulatory frameworks for various medicines, especially FMT and microbiota-modulating medications, can hinder clinical acceptance [2]. Few long-term studies have examined the efficacy and safety of these interventions in diverse patient populations [18,19,20].

This narrative review will investigate the evidence linking gut microbiota composition with atherosclerosis progression. We will explore potential pathways as well as the impact of dietary and microbial interventions. Our main aim is to understand the gut–heart axis and its implications for cardiovascular health.

## 2. Pathophysiological Mechanisms

The association between gut microbiota and atherosclerosis involves several mechanisms, particularly inflammation, lipid metabolism, and the synthesis of SCFAs [21].

### 2.1. Inflammation and Immune Activation

One of the most significant ways gut microbiota can influence atherosclerosis is through inflammatory responses. The imbalance in microbial composition can compromise gut barrier integrity. This is something called “leaky gut” (increased permeability) [22]. This mechanism allows bacterial components (LPS) to translocate from the gut into the bloodstream [23]. These potent pro-inflammatory molecules from the outer membrane of Gram-negative bacteria stimulate the immune system, triggering a systemic inflammatory response [24]. Once in circulation, LPS binds to toll-like receptor 4 (TLR4) on immune cells, activating the nuclear factor–Kappa B (NF-κB) pathway, which in turn promotes the release of inflammatory cytokines [25]. The start and progression of atherosclerosis depend on inflammation. The endothelium recruits immune cells, releasing pro-inflammatory cytokines such IL-6, TNF-α, and IL-1β, leading to plaque instability [26]. Recent studies show that gut-derived LPS and systemic inflammation interact, supporting the gut–heart axis’s role in atherosclerosis [27,28,29].

### 2.2. Lipid Metabolism Modulation

Besides inflammatory pathways, the gut microbiota profoundly affects lipid metabolism, which is a critical element in the progression of atherosclerosis. Specific gut bacteria metabolize dietary constituents, including choline, phosphatidylcholine, and L-carnitine, into trimethylamine (TMA). Figure 1 shows that TMA is converted by the liver into TMAO [23]. This compound is linked to atherogenesis, as it can strengthen cholesterol deposition in arterial walls, hinder reverse cholesterol transport, and provoke inflammatory responses in vascular tissues [30].

Wang et al. showed that elevated plasma TMAO levels are strongly linked to increased cardiovascular risk, and their study demonstrates that reducing TMA formation in the gut can attenuate atherosclerosis in mice [32]. This pathway exemplifies how the gut microbiota’s metabolic functions can directly contribute to atherogenic processes through modifications in lipid processing [33].

### 2.3. SCFA Production and Vascular Health

SCFAs, including acetate, propionate, and butyrate, are another important group of microbial metabolites that modulate host physiology [34]. Produced by gut bacteria through the fermentation of dietary fibers, SCFAs have both pro- and anti-inflammatory effects on vascular health [28]. Propionate, for instance, has anti-inflammatory properties that may protect against atherosclerosis by inhibiting vascular inflammation and reducing lipogenesis. Butyrate is known for its beneficial effects on gut health, promoting barrier integrity and anti-inflammatory cytokine production [35]. Nonetheless, the function of acetate is more intricate, as it can act as a substrate for cholesterol synthesis in the liver. This can lead to lipid accumulation in the arteries [36].

Beyond these direct impacts, gut microbiota can modulate immune cells involved in plaque formation and progression [21]. Certain bacteria produce metabolites that activate regulatory T-cells (Tregs), which have anti-inflammatory effects. They can potentially reduce plaque buildup and improve plaque stability [30]. Dysbiosis, in contrast, can result in elevated levels of Th17 cells that produce pro-inflammatory cytokines like as IL-17 [37]. They are known to accelerate atherosclerosis progression [24]. Moreover, certain bacteria such as *Streptococcus* and *Veillonella*, developed in atherosclerotic patients, have been detected within atherosclerotic plaques themselves. This means that this possible microbial translocation can directly affect plaque composition [38]. Figure 2 below summarizes some of the consequences of dysbiosis. 

Thus, the gut microbiota influences atherosclerosis through multiple pathways involving inflammatory signals, lipid metabolism, SCFA production, and immune modulation [39]. These methods show the potential of gut microbiota to act as a biomarker for cardiovascular risk and for creating therapeutic targets. It establishes the basis of the “gut–heart axis”, showing the significance of gut health in vascular health and the prevention of atherosclerosis [40].

## 3. Gut Microbiota Profiles in Atherosclerosis

The composition of gut microbiota differs significantly between individuals with atherosclerosis and healthy controls. Specific microbial profiles may influence the progression of this disease [41]. High-throughput sequencing technologies, such as 16S rRNA gene sequencing and shotgun metagenomics, have allowed researchers to identify microbial taxa associated with atherosclerotic CVD [34]. Studies on both humans and animal models have found that patients with atherosclerosis have a secondary manifestation, and that is an imbalance in gut microbial composition, with a higher prevalence of pro-inflammatory bacteria and a lower presence of beneficial, anti-inflammatory microorganisms [42,43,44]. Figure 3 shows the microbiota sites affect atherosclerosis.

One prominent finding in gut microbiota research on atherosclerosis is the increased abundance of certain bacterial genera, such as *Megamonas*, *Veillonella*, and *Streptococcus* in patients with this condition [45]. *Megamonas* has been linked to obesity and metabolic diseases, generating metabolites such as acetate and propionate that may influence lipid metabolism, Dysregulation, and inflammation [46]. Likewise, *Veillonella*, a genus proficient in amino acid fermentation, is often higher in individuals with coronary artery disease (CAD). *Veillonella* has been associated with both obesity and insulin resistance [45]. *Streptococcus* (in atherosclerotic plaques) is another genus associated with CVD [47].

Conversely, beneficial bacterial genera, such as *Bifidobacterium* and *Roseburia*, are often depleted in individuals with atherosclerosis. *Bifidobacterium* is a well-known probiotic genus that plays a role in strengthening gut barrier integrity and reducing systemic inflammation [48]. Guaman et al. have shown *Bifidobacterium* to produce acetate, which supports a healthy gut environment and reduces the translocation of endotoxins like LPS into the bloodstream [49]. In their study, they demonstrated that lower levels of *Bifidobacterium* in atherosclerosis may contribute to a leaky gut and systemic inflammation, which exacerbates cardiovascular risk [49]. *Roseburia*, acknowledged for its capacity to generate butyrate, a SCFA, is crucial for sustaining gut health by facilitating anti-inflammatory mechanisms and enhancing epithelial barrier integrity [50]. Amiri et al. reached the conclusion that the reduction of *Roseburia* in atherosclerotic patients implies a loss of protective butyrate production, and this potentially contributes to disease progression [51].

Another critical aspect of gut microbiota profiles in atherosclerosis is the *Firmicutes–to–Bacteroidetes* ratio, which is often altered in metabolic and inflammatory diseases [52]. De Filippo et al. [53] and Montenegro et al. [54] showed that increased abundance of *Firmicutes* (linked to energy metabolism and caloric absorption) and a decrease in *Bacteroidetes* (associated with anti-inflammatory effects) may contribute to a pro-inflammatory state and increased cholesterol levels, both of which are risk factors for atherosclerosis. Furthermore, according to Yoo et al. [55], specific species within these phyla, such as *Collinsella*, have been found to be more abundant in atherosclerotic patients. The authors showed that *Collinsella* is associated with increased gut permeability and inflammatory cytokine production [55].

The gut microbiota in atherosclerosis shows a unique composition characterized by a boost in pro-inflammatory, pathogenic bacteria and a decrease in beneficial, anti-inflammatory bacteria. This imbalance seems to facilitate disease progression via pathways related to inflammation, lipid metabolism, and gut barrier dysfunction [47].

## 4. Mechanisms of Dysbiosis in Atherosclerosis

Dysbiosis, the imbalance of gut microbial composition, is a significant factor in the development and progression of atherosclerosis [56]. Diet is one of the most influential factors in shaping the gut microbiome, and dietary patterns high in saturated fats, refined sugars, and low in fiber—characteristic of a Western diet—promote dysbiosis [57]. Such a diet encourages the growth of pathogenic bacteria while reducing beneficial microbes that thrive on dietary fiber, like *Bifidobacterium* and *Roseburia* [58]. The reduction of SCFA-producing bacteria in individuals with atherosclerosis contributes to a “leaky gut” (increased gut permeability), allowing pathogens to enter into the systemic circulation [59]. The translocation of microbial components, such as LPS from Gram-negative bacteria, from the gut to the bloodstream induces systemic inflammation, a main contributor to atherosclerosis [60]. Figure 4 shows the route from bacterial invasion to thrombus formation.

*Firmicutes* are more efficient at breaking down and absorbing energy from food. This contributes to obesity and metabolic syndrome, both being risk factors for atherosclerosis [61]. In contrast, *Bacteroidetes* are associated with the production of anti-inflammatory compounds, so their reduction skews the microbiome toward a more inflammatory state [62].

### 4.1. Metabolism Imbalance and Atherogenic Effects

Microbial metabolites are essential in the gut–heart axis, affecting systemic inflammation, lipid metabolism, and immunological activation [63]. Dysbiosis results in an increase of TMA-producing bacteria, such as *Escherichia* and *Klebsiella*, which consequently raises TMAO levels (chemical linked to cardiovascular risk) [64]. TMAO promotes cholesterol accumulation in macrophages, enhances foam cell formation, and induces pro-inflammatory pathways, all of which contribute to plaque development in the arteries [65]. Dysbiotic microbiomes also generate low levels of SCFAs, including butyrate and propionate, which are essential for regulating lipid metabolism and mitigating inflammation [66]. This eliminates a protective barrier against atherosclerosis [67].

### 4.2. Immune Dysregulation and Inflammation

The inflammatory environment promoted by dysbiosis extends beyond metabolites to involve direct interactions with immune cells. Koren et al. [68] detected in the atherosclerotic plaques certain bacterial genera associated with dysbiosis. In their study, the authors found that *Chryseomonas* was found in all atherosclerotic plaque samples, and that *Veillonella* and *Streptococcus* were present in the majority of samples [68]. Furthermore, the same study revealed that numerous supplementary bacterial phylotypes were prevalent in both the atherosclerotic plaque and oral or gastrointestinal samples from the same individual [68]. This translocation can promote localized inflammation, attracting immune cells and increasing the likelihood of plaque instability [69]. On this issue, Omenetti et al. mention that dysbiosis favors the expansion of Th17 cells, while reducing regulatory T cells (Tregs) that help control immune responses [64]. The authors conclude that this imbalance exacerbates systemic inflammation, worsening atherosclerosis progression [70].

### 4.3. Mechanisms of Gut Barrier Dysfunction

Dysbiosis disrupts the gut barrier, which normally prevents harmful bacteria and their metabolites from entering circulation [71]. Pathogenic bacteria like *Collinsella* prevalent in atherosclerotic patients, can modify the gut composition by boosting permeability and facilitating the production of inflammatory cytokines [72]. A weakened barrier allows bacterial compounds such as LPS and peptidoglycan to enter the bloodstream, instigating systemic inflammation and causing vascular damage. This mechanism connects the localized impacts of dysbiosis in the gastrointestinal tract to systemic inflammation and the advancement of atherosclerosis [73].

Dysbiosis in atherosclerosis is consequently induced by an imbalance of microbial species, modified metabolite synthesis, immunological dysregulation, and failure of the gut barrier. These processes jointly promote a pro-inflammatory background that accelerates plaque development and destabilization [74].

## 5. Potential of Microbiota-Based Interventions

As evidence increasingly associates gut microbiota with atherosclerosis progression, researchers are investigating microbiome-based therapies as potential therapeutic methods [75]. On this matter, Kelly et al. [76] state that these therapies can restore microbial equilibrium, diminish inflammation, and regulate the synthesis of metabolites that influence cardiovascular health. The authors also mention that main strategies incorporate the use of probiotics, prebiotics, dietary alterations, FMT and microbiota-targeted pharmacological development [76].

In a study on mice, Huang et al. [77] proved that probiotic strains, namely *Bifidobacterium* and *Lactobacillus*, had anti-inflammatory effects and the ability to enhance gut barrier integrity. In this study, *Bifidobacterium* promoted the production of acetate, which is a SCFA that reduces gut permeability and limits systemic inflammation by decreasing endotoxin translocation into the bloodstream [77]. In another study on 14 Yorkshire swine, Aboulgheit et al. demonstrated that probiotic supplementation lowers the levels of pro-inflammatory cytokines and improves lipid profiles, thus potentially reducing the risk of atherosclerosis [78]. The results reported that dietary treatment with apple sauce combined with *L. plantarum* probiotic effectively induces Nrf2 activity in vivo [78]. Although more human studies are required, probiotics represent a promising and accessible approach for managing atherosclerosis through gut microbiota modulation.

Prebiotics are non-digestible dietary fibers that stimulate the growth and activity of beneficial bacteria in the gut [79]. In a study conducted by Rossi et al. [80] that analyzed 55 bifidobacteria for the ability to ferment glucose, *Bifidobacterium* sp. ALB 1 grown on inulin showed the greatest degradative capability, fermented the longest chains of inulin, and did not exhibit stringent selectivity based on its degree of polymerization [80].

SCFAs such as butyrate, propionate, and acetate play protective roles in cardiovascular health by reducing inflammation, enhancing gut barrier integrity, and regulating lipid metabolism [81]. Increasing dietary fiber intake through prebiotics can promote the growth of these beneficial bacteria, counteracting dysbiosis associated with atherosclerosis [82]. In his study, Stojanov et al. [83] suggest that individuals with higher fiber intake have a lower Firmicutes–to–Bacteroidetes ratio. This is linked to a better metabolic health and reduced cardiovascular risk.

Diet is one of the most powerful influencers of gut microbiota composition. Ma et al. [84] associate greater microbial diversity and an increase in beneficial bacterial species with a diet high in fiber, plant-based foods, and omega-3 fatty acids. In this study, Mediterranean-style diets that are rich in fibers and polyphenols promoted SCFA production and reduced inflammation. In contrast, Aziz et al. [85] state that Western diets high in saturated fats and refined sugars promote dysbiosis by encouraging the growth of pro-inflammatory bacteria and reducing SCFA-producing microbes. 

### 5.1. Faecal Microbiota Transplantation: A Novel Approach

FMT has emerged as a promising intervention for modulating the gut microbiota in various conditions, including metabolic and cardiovascular diseases [86]. Recent studies have demonstrated its potential to improve lipid profiles, reduce systemic inflammation, and enhance gut barrier integrity [87,88,89].

FMT involves the transfer of gut microbiota from a healthy donor to a recipient, aiming to restore a balanced microbiome [90]. Although predominantly utilized for the treatment of *Clostridioides difficile* infection, FMT is being investigated for metabolic disorders, such as atherosclerosis. Early studies indicate that FMT from healthy donors can improve insulin sensitivity and reduce inflammatory markers. For example, De Groot et al. [91] noted a substantial reduction in insulin sensitivity in 22 individuals two weeks post-allogenic METS-D FMT. The authors also noted an accelerated intestinal transit time subsequent to RYGB-D FMT, alterations in faecal bile acids, inflammatory markers, and variations in several intestinal microbiota taxa.

Figure 5 below shows the impacts of dysbiosis and the role of FMT.

As understanding of the gut–heart axis progresses, researchers are exploring tailored pharmaceuticals that alter specific microbial pathways. TMA synthesis in the gut has demonstrated a potential in decreasing TMAO levels [92]. These medications aim to diminish the formation of atherogenic metabolites by specifically targeting TMA-producing bacteria, thereby preserving the total microbial diversity of the microbiome [93].

### 5.2. Summary of the Potential and Challenges of Microbiota-Based Therapies

Personalized microbiota modification may improve therapeutic effects [94]. Probiotics, prebiotics, and FMT may work best when tailored to the individual’s microbiota, genetic profile, and health status [95]. Now that microbiome sequencing and computational modelling can better profile gut microbial communities, physicians can anticipate how patients will respond to certain therapies [96].

Figure 6 shows how gut microbiota, lifestyle, and cardiovascular outcomes are linked. Diagnostic tests like microbiome profiling and microbial structural product analysis help to determine gut microbiota abundance [97]. The diagram shows lifestyle and metabolic risk factors such poor diet, lack of exercise, smoking, excessive alcohol consumption, hypertension, abnormal cholesterol levels, diabetes, and obesity.

These variables cause gut dysbiosis [98]. In combination with these risk factors, gut dysbiosis can cause cardiovascular illnesses like atherosclerosis and heart failure [99]. Pathological alterations include inflammatory immune cell infiltration and fibrosis that cause these illnesses [100,101].

## 6. Strengths, Limitations, and Future Directions

One of the strengths of this review is that it highlights how gut microbiota affects cardiovascular health and consolidates gut–heart data. Mechanistic and therapeutic approaches are uniquely integrated to provide a thorough picture of atherosclerosis therapies. Moreover, this study synthesizes recent developments and identifies important gaps, such as the long-term consequences of microbiota-targeted therapy. It outlines future studies to better understand the gut–heart axis and atherosclerosis.

This study also has some limitations that we will address here. Numerous studies examining this relationship are observational or involve animal models. This constrains the capacity to draw conclusive causal conclusions in humans. Additionally, gut microbiota composition is influenced by numerous factors such as diet, genetics, medications, and lifestyle, making it difficult to isolate specific microbial contributions to atherosclerosis.

Moreover, probiotics, prebiotics, and FMT are promising microbiota-targeting medicines, but they face many hurdles. Another limitation of this study is that it includes patient response heterogeneity due to genetic predispositions, pre-existing health problems, and gut microbiota makeup. This heterogeneity makes therapy outcome prediction and protocol creation difficult. FMT, which can transmit pathogens and cause immunological reactions, remains a safety concern.

The dependence on 16S rRNA gene sequencing and metagenomics to examine microbiota composition involves additional constraints. Although these approaches yield significant insights into microbial diversity and abundance, they inadequately address functional elements of the microbiome, including metabolic activity and interactions between microbial metabolites and host tissues. Moreover, these methods of examining the microbiota remain underutilized due to high costs and technical complexities.

Innovative microbiota-based therapies (probiotics, prebiotics, dietary alterations, and FMT) can restore microbial equilibrium by generating an abundance of anti-inflammatory, SCFA-producing bacteria while suppressing the growth of pro-inflammatory species [98].

## 7. Conclusions

The influence of gut microbiota on cardiovascular health is a groundbreaking and significant research domain, altering our comprehension of atherosclerosis as a condition affected not alone by conventional risk factors but also by the gut–heart axis. The research indicates that the composition and function of gut microbiota are closely associated with the development and progression of atherosclerosis. In this study we found that pro-inflammatory bacteria (*Megamonas* and *Streptococcus*) are linked to plaque formation and inflammatory pathways, whereas beneficial bacteria (*Bifidobacterium* and *Roseburia*) seem to provide protective mechanisms that help maintaining gut barrier integrity.

## Figures and Tables

**Figure 1 microorganisms-12-02341-f001:**
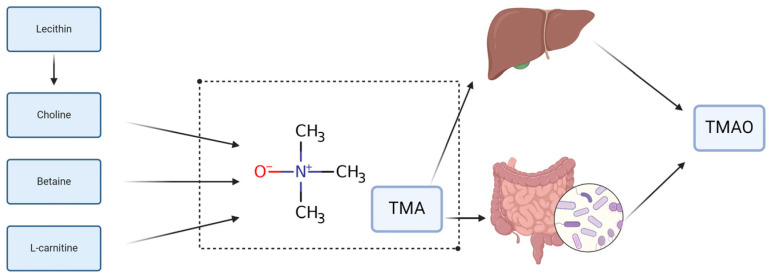
In vivo metabolic pathways of TMAO and its precursors. Choline is metabolized by the choline trimethylamine-lyase system. Betaine can be catalyzed by L-carnitine dehydrogenase and subsequently reduced to TMA. L-carnitine can be immediately turned into TMA, and TMA can be oxidized to TMAO in the liver and intestines (jejunum and cecum) before entering the bloodstream. Created with BioRender.com [31] (accessed on 29 October 2024).

**Figure 2 microorganisms-12-02341-f002:**
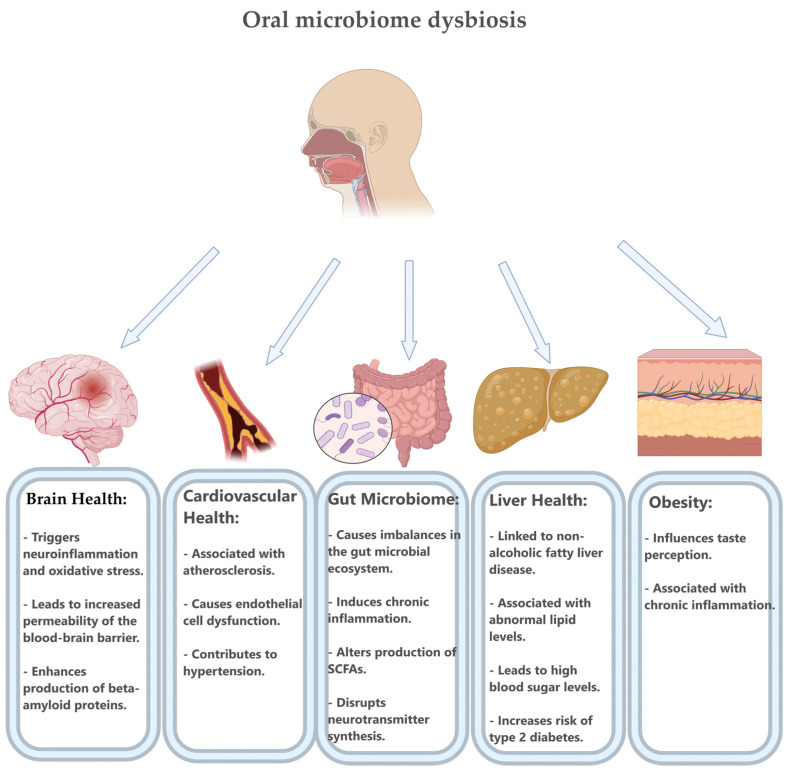
How the oral microbiome can contribute to an increased frequency of strokes. The presentation focuses on the significant adverse consequences that dysbiosis has on the central nervous system, the vascular system, the gut, the liver, the metabolism of lipids and carbohydrates, and fat tissue. Created with BioRender.com [31] (accessed on 29 October 2024).

**Figure 3 microorganisms-12-02341-f003:**
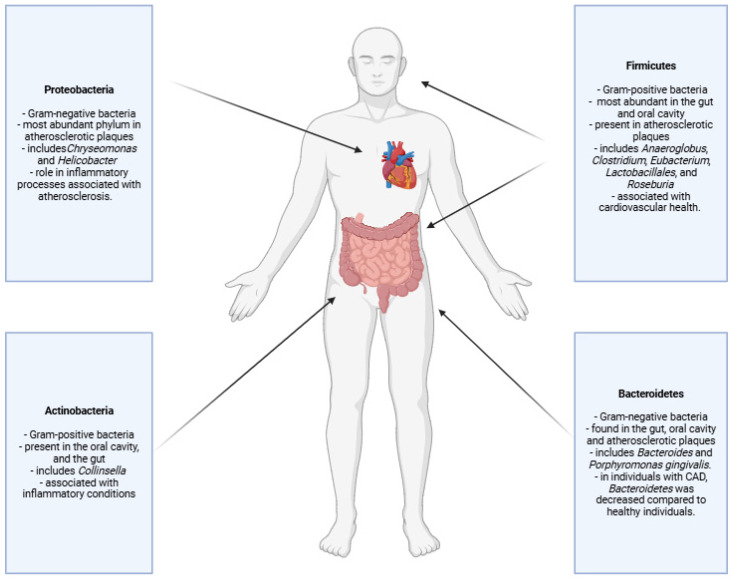
Microbiota locations that impact atherosclerosis. Bacterial DNA identified in atherosclerotic plaques may originate from bacteria located in the superior or inferior gastrointestinal tract. Created with BioRender.com (accessed on 29 October 2024).

**Figure 4 microorganisms-12-02341-f004:**
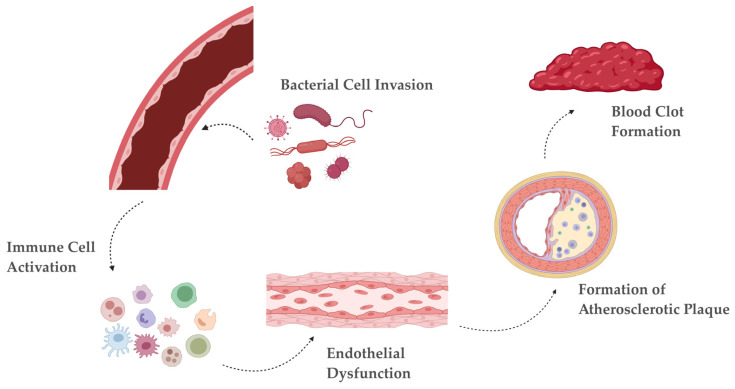
Pathway from bacterial invasion to blood clot: Bacterial cells release various components, including LPS, pathogen-associated molecular patterns (PAMPs), and outer membrane vesicles (OMVs), which can trigger an immune response. The invasion activates immune receptors such as CXCR2, CCR2, TLR-4/2, and NLRP3. This leads to an increase in inflammatory mediators, including IL-1β, IL-10, IL-17, Th17, IFN-γ, GM-CSF, G-CSF, IL-8, TNF-α, MCP1, and CRP. The immune response causes a reduction in nitric oxide (NO) levels and increases expression of endothelial adhesion molecules, such as E-selectin, ZO-1, vWF, PECAM-1, VCAM-1, and ICAM-1, disrupting endothelial function. Plaque formation involves oxidative stress, inflammatory cell infiltration, an imbalance in M1/M2 macrophage ratio, formation of foam cells, and lipid accumulation within the arterial wall. As plaque forms, factors like Cdc42 activation and increased collagen-binding proteins contribute to clotting. This leads to altered clotting and partial thromboplastin times, ultimately promoting thrombus (blood clot) formation [17]. Created with BioRender.com [31] (accessed on 29 October 2024).

**Figure 5 microorganisms-12-02341-f005:**
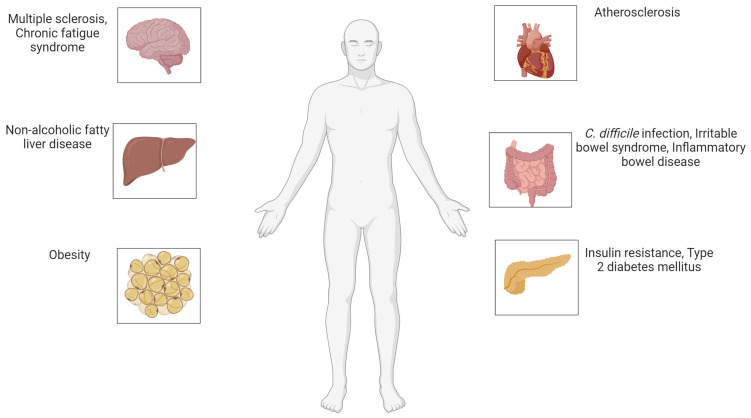
Potential health impacts of gut microbiota imbalance and role of FMT: Gut dysbiosis has been linked to neurological disorders like multiple sclerosis, cardiovascular conditions such as atherosclerosis, metabolic issues including obesity and type 2 diabetes, liver diseases like non-alcoholic fatty liver disease, and gastrointestinal disorders such as C. difficile infection, IBS, and IBD. Created with BioRender.com [31] (accessed on 29 October 2024).

**Figure 6 microorganisms-12-02341-f006:**
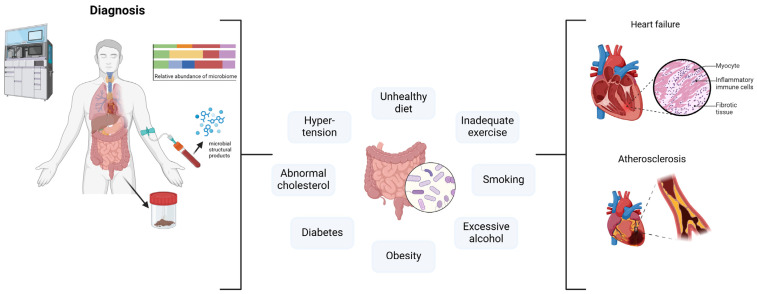
The potential and challenges of microbiota-based therapies in atherosclerosis. Created with BioRender.com [31] (accessed on 13 November 2024).

## Data Availability

No new data were created or analyzed in this study.

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
