# Peer review of "Intestinal Insights: The Gut Microbiome’s Role in Atherosclerotic Disease: A Narrative Review"

_microorganisms, 2024, doi:10.3390/microorganisms12112341_

Round 1
Reviewer 1 Report
Comments and Suggestions for Authors
This study “Intestinal Insights: The Gut Microbiome's Role in Atherosclerotic Disease” is a good read. This review examines the emerging role of gut microbiota in the development and progression of atherosclerosis, a chronic inflammatory cardiovascular disease. It discusses how microbial dysbiosis can influence host lipid metabolism, immune function, and inflammatory responses, contributing to atherogenesis. Key mechanisms include the production of pro-atherogenic metabolites like trimethylamine-N-oxide (TMAO) and short-chain fatty acids (SCFAs), as well as the translocation of inflammatory microbial components. Distinct microbiome profiles associated with atherosclerosis are highlighted, including increased pathogenic bacteria (e.g., Megamonas, Veillonella, and Streptococcus) and decreased beneficial genera (e.g., Bifidobacterium and Roseburia). These microbial shifts offer potential biomarkers and therapeutic targets. The review proposes integrating microbiota-targeting interventions—such as probiotics, prebiotics, dietary changes, and fecal microbiota transplantation—with conventional cardiovascular treatments to improve patient outcomes.
Comments:
1, Clarify the potential limitations of microbiota-targeting therapies, including variability in patient responses and potential safety concerns.
2, At least one additional Figure (illustration) may be provided as to highlight the summary or prospect of this study.
3, The English of manuscript can be polished (minor) and there are few typological errors.
4, The author should cross-check all abbreviations in the manuscript. Initially, define in full name followed by abbreviation.
5, Include the challenges of translating microbiota modulation strategies into clinical practice would be valuable.
6, Authors should add a paragraph to discuss limitations of this study.
7, Expand on how microbiota-targeting interventions, such as probiotics and FMT, could be tailored for individual patients would also enhance the therapeutic relevance.
Reviewer 2 Report
Comments and Suggestions for Authors
Review for the manuscript
Intestinal Insights: The Gut Microbiome's Role in Atherosclerotic Disease
Dear Editor,
Thank you for the invitation to review this manuscript for Microorganisms.
The above-mentioned manuscript is very interesting, and has scientific relevance. I have some minor comments and I suggest some modifications that can be seen below.
OVERALL COMMENTS
In this manuscript, the authors intended to investigate the complex mechanisms through which gut dysbiosis promotes atherogenesis. Furthermore they emphasize the potential of integrating microbiota modulation with traditional cardiovascular care, offering a holistic approach to managing of this condition that are the most cause of death worldwide. In conclusion they point that adequate interventions targeting microbiota (probiotics, prebiotics, dietary modifications, and faecal microbiota transplantation), present effective approaches for restoring microbial equilibrium and justifying cardiovascular risk.
TITLE
I suggest including the type of the study. Is it a review?
ABSTRACT
The abstract is almost adequate. Please use a block to build it and not separate paragraphs.
KEYWORDS
The keywords used were: atherosclerosis; gut metagenome; inflammation; intestinal microbiota.
I suggest including pro and prebiotics. This information is found only at the end of the Introduction section.
INTRODUCTION
I suggest including more references published in 2023 and 2024.
Ethics concerns: None, since this manuscript is a review.
OTHER SECTIONS
In the section: 2.1. Inflammation and Immune Activation
Inflammatory process is crucial for cardiovascular diseases occurrence. There is much more information regarding inflammation and atherosclerosis. I suggest expanding this section.
In Figure 3, I suggest reducing the information found in each “balloon”.
Please, be sure that all the acronyns used in the text were defined.
Please check if the names of all microorganisms are written in italics. See as example the legend of Figure 3 where C. difficile is not.
In line 292 we can read that “Prebiotics are non-digestible dietary fibers that stimulate the growth and activity of beneficial bacteria in the gut [73]”. The words “prebiotic” and probiotic appeared before in the text. Please define the first time you cited.
I also believe that the authros could expand the section “Fecal Microbiota Transplantation”. There are many brand new references regarding this topic.
CONCLUSIONS
This section is adequate for the study findings. However, at the end of this section we can read that “Innovative microbiota-based therapies (probiotics, prebiotics, dietary alterations, and FMT) can restore microbial equilibrium by generating an abundance of anti-inflammatory, SCFA-producing bacteria while suppressing the growth of pro-inflammatory species.”
This sentence fits better in “Future perspectives”
LIMITATIONS AND FUTURE DIRECTIONS
I appreciate the inclusion of limitations of this review.
However, this review also has strengths. Please include these strenghs.
REFERENCES
As pointed out above, I suggest including more references published in 2023 and 2024 in the Introduction section.
Reviewer 3 Report
Comments and Suggestions for Authors
Intestinal Insights: The Gut Microbiome's Role in Atherosclerotic Disease
The gut microbiota, consisting of a diverse collection of microorganisms, impacts host metabolism, immune responses, and lipid processing, all of which contribute to atherosclerosis. This review explores the complex mechanisms through which gut dysbiosis promotes atherogenesis. The authors emphasized on the potential of integrating microbiota modulation with traditional cardiovascular care, offering a holistic approach to managing atherosclerosis.
Recent advances have highlighted the gut microbiota as a significant contributor to the development and progression of atherosclerosis, which is an inflammatory cardiovascular disease (CVD) characterized by plaque buildup within arterial walls.
The review is well prepared, well organized, well written
There are few minor points which will improve the presentation
Revise and organize the abstract to reflect the topics you discussed in the review
Figure numbers are not correct. I guess you should try to reproduce some of those figures. Sometimes, the location of the figure is not in the correct place in the review.
In some locations, you are missing citation. For example, L141.
Round 2
Reviewer 2 Report
Comments and Suggestions for Authors
Dear authors,
Thank you very much for addressing my suggestions.
With best regards,
Dr Sandra M. Barbalho